# Definition of a New HLA B*52-Restricted Rev CTL Epitope Targeted by an HIV-1-Infected Controller

**DOI:** 10.3390/v15020567

**Published:** 2023-02-18

**Authors:** Boutaina El Kenz, Katja G. Schmidt, Victoria K. Ogungbemi-Alt, Silke Bergmann, Philipp Steininger, Klaus Korn, Bernd Spriewald, Ellen G. Harrer, Krystelle Nganou-Makamdop, Thomas Harrer

**Affiliations:** 1Infectious Diseases and Immunodeficiency Section, Department of Internal Medicine 3, Universitätsklinikum Erlangen, Friedrich-Alexander-Universität Erlangen-Nürnberg, 91054 Erlangen, Germany; 2Institute of Clinical and Molecular Virology, Universitätsklinikum Erlangen, Friedrich-Alexander-Universität Erlangen-Nürnberg, 91054 Erlangen, Germany; 3Department of Internal Medicine 5, Universitätsklinikum Erlangen, Friedrich-Alexander-Universität Erlangen-Nürnberg, 91054 Erlangen, Germany

**Keywords:** HIV-1 infection, elite controller, CTL, epitope, Rev, HLA B*52

## Abstract

The analysis of T-cell responses in HIV-1-infected controllers may contribute to a better understanding of the protective components of the immune system. Here, we analyzed the HIV-1-specific T-cell response in a 59-year-old HIV-1-infected controller, infected for at least seven years, who presented with low viral loads ranging from <20 copies/mL to 200 copies/mL and normal CD4 counts of >800 cells/µL. In γ-IFN-ELISpot assays using freshly isolated PBMCs, he displayed a very strong polyclonal T-cell response to eight epitopes in Gag, Nef and Rev; with the dominant responses directed against the HLA-B*57-epitope AISPRTLNAW and against a so-far-unknown epitope within Rev. Further analyses using peptide-stimulated T-cell lines in γ-IFN-ELISpot assays delineated the peptide RQRQIRSI (Rev-RI8) as a newly defined HLA-B*52-restricted epitope located within a functionally important region of Rev. Peptide-stimulation assays in 15 HLA-B*52-positive HIV-1-infected subjects, including the controller, demonstrated recognition of the Rev-RI8 epitope in 6/15 subjects. CD4 counts before the start of antiviral therapy were significantly higher in subjects with recognition of the Rev-RI8 epitope. Targeting of the Rev-RI8 epitope in Rev by CTL could contribute to the positive association of HLA-B*52 with a more favorable course of HIV-1-infection.

## 1. Introduction

HIV-1-specific cytotoxic T-lymphocytes (CTL) play an important role in the control of HIV-1. The emergence of CTL was associated with suppression of HIV-1 replication at primary HIV-1-infection [1] and the frequency of HIV-1-specific CTL correlated to a lower plasma viremia in the chronic phase of HIV-1-infection [2]. However, despite frequent detection of HIV-1-specific CTL in the asymptomatic stage of HIV-1 infection, without antiviral therapy most HIV-1-infected patients develop progressive immunodeficiency. Only a small group of subjects, the long-term non-progressors, maintain normal CD4 counts and low viral loads for more than ten years, and only a few subjects, the elite controllers, are able to suppress viral replication below 50 copies/mL [3]. The reasons for the long-term control of HIV-1-replication are not fully defined and may vary between individual controllers. Strong HIV-1-specific CTL responses have been observed in at least a subgroup of long-term non-progressors (4). The importance of CTL-mediated control is underlined by the strong correlation of certain HLA-I alleles, such as HLA B*57, B*27, B*52 or B*14, with a more favorable course of HIV-1 infection [4,5,6]. It has been shown that the protective effect of these alleles depends on the recognition of specific CTL epitopes located in conserved areas of the HIV-1 genome, predominantly in Gag [7,8,9]. However, HIV-1 can evade CTL recognition by development of escape mutations in CTL epitopes decreasing either processing or binding of the viral peptide to the HLA molecule or the recognition of the peptide by the T-cell receptor [10]. Mutations within protective CTL epitopes usually have a strong negative impact on viral replication, however, HIV-1 can eventually counteract the detrimental effects of such mutations by introduction of compensatory mutations outside of the epitope [11,12,13]. For the development of effective vaccines, it is important to understand why only a minority of subjects, targeting known protective epitopes, are able to avoid immune escape and to achieve long-term control of HIV-1-replication. The analysis of the CTL response in these HIV-1-infected elite controllers provides an important opportunity to delineate the contribution of targeting specific CTL epitopes for an efficient suppression of HIV-1-replication. Here, we report on a HIV-1-infected elite controller displaying a very strong polyclonal CTL response against at least eight epitopes. In addition to his dominant CTL responses against two known HLA B*57-restricted Gag epitopes, he showed a similarly strong response against a newly defined HLA B*52-restricted epitope in Rev. Our results indicate the importance of targeting multiple epitopes to avoid CTL escape. Furthermore, recognition of this HLA B*52-restricted Rev epitope could contribute to the association of HLA B*52 with a more benign course of HIV-1-infection.

## 2. Materials and Methods

### 2.1. Study Subjects

The 59- year-old controller (subject #1) was diagnosed with HIV-1 infection when he was treated in the hospital for recurrent epileptic seizures. Since an ischemic stroke seven years previously he had suffered from hemiparesis of his right leg and right arm. He was a smoker with hypertonic blood pressure. The time point of HIV-1 infection is not known but was established to have occurred prior to his severe neurological disability.

At diagnosis, he displayed a CD4 count of 1004 cells/µL and HIV-1-specific antibodies measured by ELISA (ARCHITECT HIV Ag/Ab Combo Assay, Abbott, Wiesbaden, Germany) and immunoblot (Geenius HIV 1/2 Confirmatory Assay, Bio-Rad laboratories, Feldkirchen, Germany). He maintained normal CD4 counts >800 cells/µL over the next 453 days post-diagnosis (Table 1). His viral load measured by real-time HIV-1 PCR (Abbott RealTime HIV-1 assay, Abbott, Wiesbaden) was 40 copies/mL at diagnosis. with subsequent low viral loads ranging between <20 and 20 copies/mL until day 293 post-diagnosis (Table 1). At day 383, four weeks after a traumatic subarachnoid hemorrhage and fracture of his right humerus, he displayed a transient increase of viral load to 250 copies/mL with spontaneous decline to <20 copies/mL at day 453. A resistance analysis from plasma obtained at day 383 post-diagnosis revealed the presence of several mutations in reverse transcriptase (RT, 41L, 210W, 215A) and protease (33F, 43T, 46L, 53L, 82A, 88D) which were associated with high-level resistance against zidovudine, stavudine and several protease inhibitors. This indicated the transmission of a drug-resistant virus as the patient has not been treated with antiretroviral drugs in the past.

At the time point of his first viral load measurement, he was treated with the following drugs: lamotrigine, levetiracetam, lacosamide, simvastatin, acetylsalicylic acid, ramipril, amlodipine, melperone, baclofen, citalopram and thiamine. The patient’s HLA-type was HLA A*11, B*52, B*57, C*6, C*12.

In addition to the controller (subject #1), we investigated 14 HLA-B*52-positive, HIV-1-infected patients (clinical characteristics shown in Appendix A). All were on antiretroviral combination therapy (cART) for a median time of 75 months (range 3–315 months). They presented with a current median viral load of <20 copies/mL (range: <20 to 40 copies/mL) and a current median CD4 count of 872 (range 351–1434).

### 2.2. Peptides

Synthetic crude peptides were produced by EMC Microcollections (Tübingen, Germany) as C-terminal carboxamides with >70% purity of confirmed by ESI-LCMS (EMC Microcollections, Tübingen, Germany). Peptides were dissolved in H_2_O with 10% DMSO (Merck Millipore, Darmstadt, Germany) and 1% DTT (Sigma-Aldrich, Steinheim, Germany). Complete sets of HIV-1 Consensus B peptides spanning Gag (ARP-8117), Nef (ARP-5189), Rev (ARP-6445) and VPU (ARP-6444) were generously provided by the NIH HIV Reagent Program, Division of AIDS, NIAID, NIH. Gag pool 9 comprised the following five peptides overlapping in 11 amino acids: EKAFSPEVIPMFSAL, SPEVIPMFSALSEGA, IPMFSALSEGATPQD, SALSEGATPQDLNTM and EGATPQDLNTMLNTV

### 2.3. Isolation of PBMCs

PBMCs were isolated from HIV-1-infected patients by density-gradient centrifugation, in Leucosep tubes (Greiner Bio One GmbH, Frickenhausen, Germany) pre-filled with 15 mL of Lymphoflot separation medium (Bio-Rad Laboratories GmbH, Feldkirchen, Germany). The tubes were filled with a maximum volume of 30 mL citrate blood diluted with PBS to 35 mL, then centrifuged 20 min at 760× *g* and RT. The supernatant was collected and centrifuged 10 min at 610× *g* and RT. The resulting cell pellet was washed once in 30 mL PBS and then resuspended in appropriate culture medium.

### 2.4. Generation of B-Lymphoblastoid Cell Lines (B-LCL)

Next, 10 × 10^6^ freshly isolated PBMCs were resuspended in 4 mL sterile filtered supernatant from the EBV-producing B95-8 cell line, treated with cyclosporine at a final concentration of 10 µg/mL and transferred to T25 flasks. R20 medium consisting of RPMI 1640 medium (Sigma-Aldrich, Steinheim, Germany) with 20% heat-inactivated fetal bovine serum (FCS) (PAN Biotech GmbH, Aidenbach, Germany), 1% L-glutamine (2 mmol/L), penicillin (100 U/mL), streptomycin (100 mg/mL), Hepes (10 mmol/L) (Merck KGAA, Darmstadt, Germany) and fresh cyclosporine were added to cell culture after 2–3 days.

### 2.5. Generation of HIV-1-Specific T-Cell Lines

We then stimulated 5 × 10^6^ to 10 × 10^6^ PBMCs with peptides at a final concentration of 4 µg/mL in 1 mL R10-IL2 medium consisting of RPMI 1640 medium (Sigma-Aldrich, Steinheim, Germany) with 10% heat-inactivated fetal bovine serum (FCS) (PAN Biotech GmbH, Aidenbach, Germany), 1% L-glutamine (2 mmol/L), penicillin (100 U/mL), streptomycin (100 mg/mL), Hepes (10 mmol/L) (Merck KGAA, Darmstadt, Germany) and 1000 U/mL recombinant interleukin-2 (IL-2) (Proleukin, Chiron, CA, USA). After 10–14 days, outgrowing cells were tested for recognition of peptides by γ-Interferon Enzyme-Linked Immunospot Assays (γ-IFN-ELISpot).

### 2.6. Detection of HIV-1-Specific T-Cells

The screening of peptide-reactive T cells was assessed by measuring γ-INF production upon PBMC stimulation in ELISpot assays, using R5AB medium consisting of RPMI 1640 medium with 5% heat-inactivated human AB-Serum (Sigma-Aldrich, Steinheim, Germany) and the supplements 1% L-glutamine (2 mmol/L), penicillin (100 U/mL), streptomycin (100 mg/mL) and Hepes (10 mmol/L) (Merck KGAA). Then, 96-well plates (MAIPS 4510, Millipore) were activated with 25 µL/well 70% methanol, washed twice with 200 µL PBS, coated with 50 µL/well anti-human γ-interferon antibody 1-D1K (Mabtech, Stockholm, Sweden) at a final concentration of 10 µg/mL and incubated at 4 °C overnight. The plates were then washed ×4 with PBS and blocked with 150 µL/well R5AB medium.

For ELISpot culture assay, 50 µL of T-cell suspension was added to 50 µL R5AB medium in each well of a coated plate. Then, 2 µL of dissolved peptides were added directly into the wells to a final concentration of 20 µg/mL. Respective cell suspensions without peptides served as negative controls. The plates were incubated for 18 to 40 h at 37 °C and 7% CO_2_.

For color development, the plates were first washed six times with PBS containing 0.05% Tween-20 (PBS-T0.05%), then incubated with 100 µL/well biotinylated anti-human γ-IFN monoclonal antibody 7-B6-1 (Mabtech, Stockholm, Sweden) at a final concentration of 2 μg/mL for two hours at room temperature. Afterwards, the plates were washed again with PBS-T0.05% six times, then incubated with 100 μL/well avidin/peroxidase substrate (Vectastain^®^ Elite^®^ ABC-Kit, Linaris, Wertheim, Germany) for one to three hours at room temperature, then washed again three times with PBS-T0.05% and three times with PBS. Finally, 100 μL AEC substrate (3-amino-9-ethylcarbazole, Sigma-Aldrich, Steinheim, Germany) containing 0.06% H_2_O_2_ was added as chromogen to each well. The spots developed within ten minutes, and the reaction was stopped by washing the plates three times with distilled water. The plates were air-dried, and the spots were counted using an ELISpot reader (AID, Strassberg, Germany). Results are reported as SFUs (spot-forming units) either per 50 μL or per cell number.

A peptide-specific response was defined as positive if the number of spot-forming units (SFUs) exceeded the following thresholds: ≥10 SFUs and >2 fold over background (SFUs without peptide).

To assess the functional avidity of peptides by peptide titration assay, peptides were added in serial dilutions ranging from 20 µg/mL to 1 ng/mL to ELISpot plates and incubated with T-cell lines (1 × 10^5^ per well in duplicates) for 22–36 h.

### 2.7. HLA Typing

HLA class-I typing was performed using standard serological techniques (Biotest AG, Dreieich, Germany) or genotypic analyses (enzyme-linked probe hybridization assay Biotest ELPHA, Biotest AG) according to the manufacturer’s guidelines.

### 2.8. HLA-Restriction Analysis

HLA-I-/II-restriction was determined by ELISpot assays using antibodies blocking either CD4 or CD8. The cells were incubated at a final concentration of 50 μg/mL of an anti-CD4 (OriGene Technologies, Rockville, Maryland, USA) or 625 ng/mL of an anti-CD8 antibody (BD Bioscience, San Jose, CA, USA) for one hour at 37 °C and 7% CO_2_. Then, peptides were added directly to the wells at a final concentration of 20 μg/mL. The restricting HLA-I allele was determined using HLA-matched allogeneic B-LCLs from HLA-I-typed blood donors. These target cells were incubated with the respective peptides for one hour and washed twice with 15 mL PBS. Peptide-sensitized target cells and target cells without peptides were then co-incubated with the T-cell lines in a γ-IFN ELISpot assay, using 2 × 10^5^ target cells per well and 2 × 10^5^ T-cells in duplicate.

### 2.9. Statistical Analysis

The statistical analyses were carried out using GraphPad Prism 8.0. (GraphPad Soft-162 ware, San Diego, CA, USA).

## 3. Results

### 3.1. Detection of a Strong Polyclonal T-Cell Response in Controller #1

We investigated HIV-1-specific T-cell responses in freshly isolated PBMCs from subject #1 to a broad range of peptides spanning HIV-1 proteins Gag, Nef, Rev, Vpu and RT in a γ-IFN-ELISpot assay (Figure 1a). The patient displayed a very strong polyclonal T-cell response to eight individual peptides in Gag, Nef and Rev comprising seven known CTL epitopes. Four peptides were recognized with a very high frequency of >100 SFUs/200,000 PBMCs. The highest T-cell-magnitude was elicited by the HLA-B*57-restricted peptide Gag-AW10 (AISPRTLNAW) (473 SFUs/2 × 10^5^ PBMCs), followed by peptide Rev07-11 (RRRRWRERQRQIRSI, 329 SFUs/2 × 10^5^ PBMCs), the Gag-pool 9 (containing five overlapping 11 amino acid long peptides, sequences given in Section 2.2) comprising the HLA-B*57 epitope KAFSPEVIPMF (244 SFUs/2 × 10^5^ PBMCs), the Gag peptide Gag-AL15 ACQGVGGPGHKARVL (238 SFUs/2 × 10^5^ PBMCs) containing the known HLA-A*11-restricted CTL epitope Gag-CK10 CQGVGGPGHK, and the HLA-B*57-restricted peptide Nef-HW9 (HTQGYFPDW, 190 SFUs/2 × 10^5^ PBMCs). In addition, we detected moderate recognition of the HLA-A*11-restricted epitopes Nef-QK10 QVPLRPMTYK (22 SFUs/2 × 10^5^ PBMCs) and Gag-AK9 ATLYCVHQK (50 SFUs/2 × 10^5^ PBMCs (Figure 1a). Confirmation of peptide recognition and further mapping of recognized T-cell epitopes was performed in peptide-stimulation assays (Figure 1b). For this purpose, we generated T-cell lines in-vitro by stimulation of PBMCs with the seven peptides which were recognized in the ELISpot assays using freshly isolated PBMCs. In addition, we included the HLA-B*57-restricted epitope Gag-TW10 TSTLQEQIGW. Outgrowing cells were tested after 10–14 days for recognition of respective peptides in γ-INF ELISpot assay (Figure 1b). The highest CTL frequencies were evoked by the peptides AW10 (996 SFUs), KF11 (995 SFUs), Nef-HW9 (944 SFUs) and Rev07-11 (880 SFUs). In addition, the Gag peptides AK9 ATLYCVHQK, TW10 TSTLQEQIGW and CK10 CQGVGGPGHK were also well recognized but at lower frequencies (343 SFUs, 288 SFUs and 266 SFUs, respectively). Overall, the HIV-1 controller #1 showed strong responses to several known CTL epitopes as well as a strong response to the Rev peptide Rev07-11 for which no known CTL epitope has yet been described [14].

### 3.2. Definition of the Rev-RI8 Peptide as a New Epitope in Rev

To define the minimal epitope within Rev07-11, peptide-stimulated T-cell lines were analyzed for recognition of 15 amino acid long peptides overlapping with the Rev07-11 by 11 amino acids. Recognition of the peptides Rev07-11 (RRRRWRERQRQIRSI and Rev07-12 (WRERQRQIRSISGWI) with lack of recognition of the peptides Rev07-13 (QRQIRSISGWILSTY) and Rev07-10 (ARRNRRRRWRERQRQ) indicated that the epitope was located with the sequence WRERQRQIRSI (Figure 2a). The fine specificity of CTL was then mapped by a set of overlapping short peptides of 8 to 10 amino acids revealing the peptide Rev-RI8 (RQRQIRSI) as the minimal epitope (Figure 2b). Omission of the arginine at the first position or the isoleucine at the C-terminal end abrogated recognition (Figure 2b).

### 3.3. Peptide Rev-RI8 Is Presented by HLA B*52

To confirm that the observed γ-IFN responses to the rev peptide Rev07-11 with minimal epitope Rev-RI8 are mediated by CD8+ T-cells and not by CD4+ T-cells, we next performed a peptide-recognition assay using the controller’s T-cell lines in the presence of anti-CD4 or anti-CD8 antibodies. Recognition of peptide Rev-RI8 in γ-IFN ELISpot assays could be blocked by anti-CD8-antibodies but not by anti-CD4-antibodies, demonstrating presentation of the Rev-RI8 epitope by HLA-I molecules (Figure 3a). HLA-restriction of Rev-RI8 was analyzed in γ-IFN-ELISpot assays using Rev-RI8 -specific CTL lines and peptide-sensitized allogenic EBV generated B-LCL sharing only one HLA-1-allele each with controller #1. Strong recognition of the HLA-B*52-positive B-LCL loaded with Rev-RI8 peptide (SFUs 203) demonstrated presentation of Rev-RI8 peptide by HLA-B*52 (Figure 3b).

The sensitizing activity of peptide Rev-RI8 was determined by incubation of serial dilutions of the Rev-RI8 peptide with the Rev-RI8-specific T-cell lines from three HLA-B*52-positive subjects in IFN-γ ELISpot assays. The 50% peptide-sensitizing activity was determined as 2.3 µg/mL with loss of peptide recognition at 10 ng/mL (Figure 4).

### 3.4. Recognition of Rev-RI8 in a Cohort of HLA-B*52-Positive HIV-1-Infected Patients

To investigate how frequently the Rev-RI8 epitope is targeted by HLA-B*52-positive HIV-1-infected patients, recognition of the Rev-RI8 was assessed in an additional 14 HLA-B*52-positive HIV-1-infected subjects of the Erlangen HIV cohort. As all patients were on an effective antiretroviral therapy, which is associated with a decline of circulating HIV-1-specific CTL, the prevalence of Rev-RI8-specific T-cells was assessed by stimulation of PBMCs with peptide Rev-RI8 and analysis of outgrowing cells after 10–14 days in γ-IFN -ELISpot assays. Rev-RI8-specific T-cells could be detected in 6 out of 15 HLA B*52-positive patients (40%) including controller #1 (Figure 5, Appendix A). Subjects with recognition of Rev-RI8 displayed significantly higher pre-ART CD4+ T-cells (Figure 6, *p* = 0.012), whereas there were no differences regarding current CD4 counts, current and pre-ART viral loads, current and pre-ART CD8 counts, or time on antiretroviral therapy (Appendix A). The higher pre-ART CD4 counts of Rev-RI8 responders was significant, even when omitting controller #1 from the analysis (*p* = 0.0290).

## 4. Discussion

In this study, we identified an HIV-1-infected controller with a very strong polyclonal CTL response targeting at least eight different CTL epitopes including a newly defined HLA-B*52-restricted CTL epitope in Rev. Strong CTL responses have previously been demonstrated at least in subsets of HIV-1-infected controllers supporting an important role of CTL in the control of HIV-1 [15]. The HLA-B*57+ controller #1 recognized at least four HLA B*57-restricted CTL epitopes, three in Gag and one in Nef, with the highest CTL frequency targeting the HLA-B*57-restricted Gag epitope AW10. This is in line with reports by other groups that HLA-B*57 is associated with a better suppression of HIV-1 replication and a more favorable course of HIV-1-infection [4,6,16,17]. In HLA B*57+ subjects, the Gag epitopes KF11, AW10 and TW10 are usually immunodominant epitopes targeted by higher CTL frequencies than other epitopes restricted by other HLA alleles [4].

Interestingly, we observed in controller #1 very high CTL frequencies targeting the newly defined HLA-B*52-restricted CTL epitope Rev-RI8 in Rev at a similar magnitude as the immunodominant HLA B*57-epitope AW10. With a glutamine at the P2 position and an isoleucine at the C-terminal position, the Rev-RI8 epitope corresponds to the published peptide-binding motif of HLA B*52-restricted epitopes [18]. While there is sequence variation within the epitope, the P2 and P8 anchor positions are highly conserved (Appendix A) [19]. HLA-B*52 has been associated with resistance to HIV-1-infection [20], with early control after infection [16] and with non-progression to AIDS [21,22,23,24,25]. So far, it has been speculated that this beneficial effect of HLA-B*52 is mediated by CTL selecting escape mutations within HLA B*52-restricted Gag- and Pol-epitopes leading to decreased viral replicative capacity [22,26]. Here, we report on the delineation of the Rev-RI8 peptide as a new HLA-B*52-restricted CTL epitope in Rev. The Rev-RI8 epitope is located in a functionally important sequence region in Rev [27]. The first three amino acids of the Rev-RI8 epitope are part of a sequence domain that mediates both the nuclear localization and the binding of Rev to the Rev-responding element. The amino acids at positions P5 to P8 are required for multimerization of Rev. Thus, we could hypothesize that CTL escape mutations within Rev-RI8 could affect Rev function. However, due to the low viral load, we were not able to obtain autologous Rev sequences from controller #1.

Recognition of the Rev-RI8 epitope was significantly associated with higher pre-ART CD4 counts in our cohort of HLA-B*52+ patients indicating a beneficial role of Rev-RI8-specific CTL. This is in line with another study reporting on a slower progression to AIDS in subjects with detection of Rev-specific CTL [24]. In contrast, two other studies did not find a statistical correlation between the magnitude or breadth of Rev-specific CTL and the control of HIV-1-replication [28,29], but HLA-B*52 positivity and HLA-B*52-specific CTL responses were not reported in both studies. This suggests that beneficial effects of Rev-specific CTL do not depend on recognition of Rev per se, but on targeting mutationally constrained CTL epitopes in which escape mutations lead to a lower fitness of the virus [8]. To understand the relative contribution of recognition of the Rev-RI8 epitope for the control of HIV-1, further studies are needed to analyze the effect of mutations within the Rev-RI8 epitope both on CTL recognition and on Rev function.

The patient displayed several drug-resistance mutations within RT and protease demonstrating infection with a drug-resistant virus as the patient had never received antiretroviral therapy. As several of the observed drug-resistance mutations could impair viral replication capacity [30,31], we hypothesize that the transmission of an attenuated drug-resistant virus could have contributed to the long-term control of HIV-1 in this controller. This is supported by the observations of another group reporting on frequent detection of drug-resistant viruses in plasma viral sequences in individuals who became HIV controllers [32].

It has been shown that Gag-specific CTL epitopes are frequently immunodominant and that targeting Gag was associated with control [33,34]. Furthermore, it has been speculated that this beneficial role of Gag responses could be attributed to high abundance of Gag and Gag-derived peptides in infected cells [35,36], to mutational constraints preventing rapid CTL escape [8,9] and the ability of Gag-specific CD8+ T cells to target incoming virions, thus being able to eliminate infected cells prior to production of virions [37]. Although the HLA-B*57 epitopes in Gag are conserved, HIV-1 can develop escape mutations within these epitopes [38,39] and compensate the fitness costs of these mutations by introduction of compensatory mutations outside of the epitopes [40,41]. Therefore, targeting of multiple CTL epitopes in functionally important viral sequences seems to be decisive to lower the risk of immune evasion by escape mutations [42]. In this context, it is remarkable that the controller targeted the HLA-B*52-restricted Rev-RI8 epitope with similar CTL frequency as the HLA-B*57-restricted Gag-epitopes suggesting a highly efficient presentation of the Rev-RI8 epitope. As Rev is expressed early in the viral replication cycle, recognition of this immunogenic Rev epitope could have been an important factor in the long-term control of HIV-1 in this controller. Further studies are needed to investigate whether inclusion of the Rev-RI8 Rev epitope into vaccines could improve CTL-mediated control of HIV-1 in HLA-B*52-positive subjects.

## Figures and Tables

**Figure 1 viruses-15-00567-f001:**
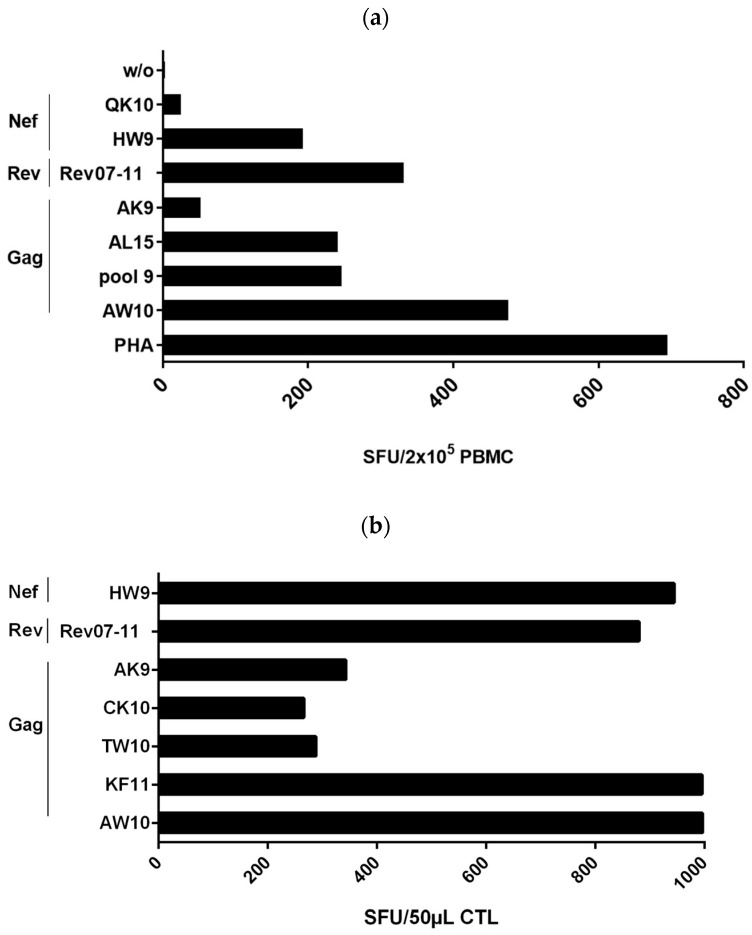
(**a**) Recognition of HLA-matching HIV-1 peptides by freshly isolated PBMCs from controller #1. A total of 2 × 10^5^ PBMCs were incubated directly after isolation for 40 h with peptides corresponding to HLA-I-restricted epitopes in γ-IFN ELISpot assay at a final concentration of 20 µg/mL. The mitogen phytohaemagglutinin (PHA) served as a positive control. w/o: PBMCs without peptide. (**b**) Recognition of HIV-1 peptides by specific T-cell lines from controller #1. PBMCs were incubated with peptides corresponding to HLA-I-restricted epitopes and IL2. Ten to fourteen days after stimulation, 50 μL of outgrowing CTL were tested in γ-IFN ELISpot assay for peptide recognition at a final concentration of 20 μg/mL Peptide-specific SFUs, after subtraction of the background without peptide, are presented. Peptides: Nef-QK10: QVPLRPMTYK, Nef-HW9: HTQGYFPDW, Rev07-11: RRRRWRERQRQIRSI, Gag-AK9: ATLYCVHQK, Gag-AL15: ACQGVGGPGHKARVL, Gag-pool 9 (containing 5 overlapping 11 amino acid long peptides including peptide EKAFSPEVIPMFSAL, further sequences given in Section 2.2), Gag-AW10: AISPRTLNAW, Gag-CK10 CQGVGGPGHK, Gag-TW10: TSTLQEQIGW, Gag-KF11: KAFSPEVIPMF.

**Figure 2 viruses-15-00567-f002:**
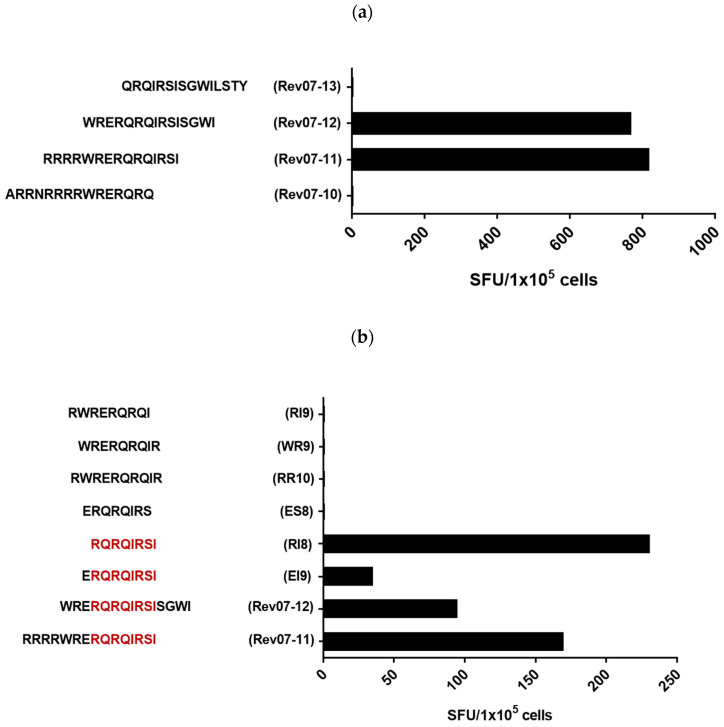
Mapping of the minimal epitope targeted by Rev-07–11-specific T-cells in controller #1. PBMCs were stimulated with Rev-07-11 peptide and IL-2 and outgrowing cells were tested after 10–14 days for recognition of overlapping and truncated peptides at a final peptide concentration of 20 µg/mL in γ-IFN-ELISpot assays. Shown are spot-forming units (SFUs) per 1:10^5^ cells. Peptide-stimulated T-cell lines without peptides served as negative controls. (**a**) Analysis of recognition of the overlapping 15-mer peptides Rev07-10, Rev07-11, Rev07-12 and Rev07-13. (**b**) Fine mapping of the epitope with truncated peptides delineated Rev-RI8 as the minimal epitope (marked in red).

**Figure 3 viruses-15-00567-f003:**
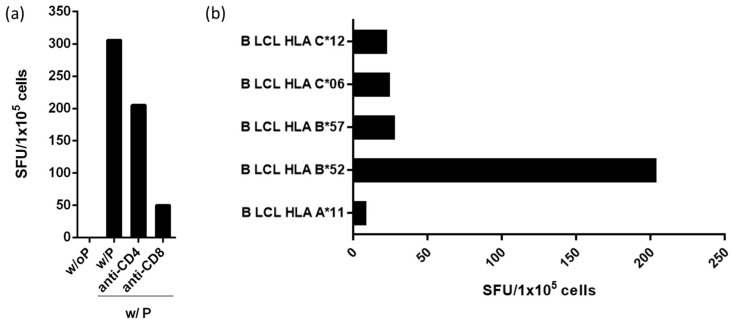
Analysis of HLA-restriction of the Rev epitope Rev-RI8. (**a**) A total of 1 × 10^5^ cells of the Rev-RI8-specific T-cell line from controller #1 were incubated in a γ-IFN ELISpot assay with peptide Rev-RI8 and blocking anti-CD4- or anti-CD8-antibodies. w/op: cells without peptide; w/P: cells with peptide Rev-RI8 (**b**) A total of 1 × 10^5^ cells of the Rev-RI8-specific T-cell line from controller #1 were incubated in a γ-IFN ELISpot assay with 1 × 10^5^ allogeneic Rev-RI8-peptide-loaded B-LCLs sharing single HLA alleles with controller #1. Shown are spot-forming units (SFUs) after incubation of T-cells with Rev-RI8-loaded B-LCL after subtraction of background, which was assessed by co-incubating T-cell lines with B-LCLs without addition of the peptide. Shown are the HLA alleles of the B-LCLs shared by the T-cell line (HLA-I type of controller #1 (A*11, B*057, B*52, C*06, C*12).

**Figure 4 viruses-15-00567-f004:**
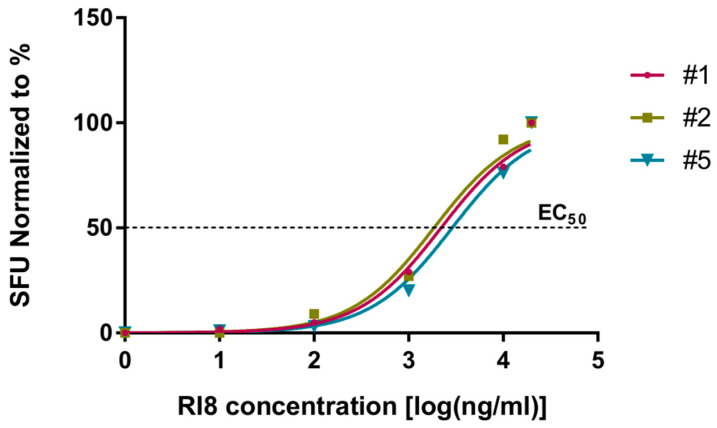
Analysis of peptide-sensitizing capacity of peptide Rev-RI8. A total of 1 × 10^5^ cells each from Rev-RI8-specific T-cell lines from the three HLA-B*52-positive subjects, #1, #2 and #5, were incubated with serial dilutions of peptide Rev-RI8 in γ-IFN ELISpot assays. Shown are adjusted spot-forming units (SFUs) after subtraction of the background. In this graph, 100 % represents the highest frequencies of SFUs determined for the individual T-cell line. The 50% peptide sensitizing activity (EC_50_) was calculated using non-linear curve regression analysis. EC_50_ values: subject #1: 2.2 µg/mL, subject #2: 1.8 µg/mL, subject #5: 2.9 µg/mL; mean: 2.3 µg/mL.

**Figure 5 viruses-15-00567-f005:**
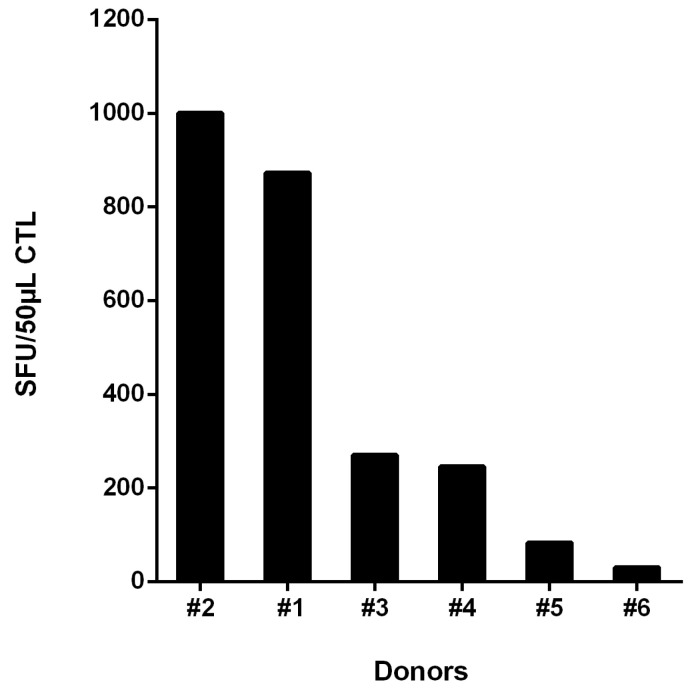
Recognition of peptide Rev-RI8 by HLA-B*52-positive HIV-1-infected subjects. PBMCs were incubated with peptide Rev-RI8 and IL2. After 10–14 days, 50 μL of outgrowing CTL were tested for peptide recognition at a final peptide concentration of 20 μg/mL.

**Figure 6 viruses-15-00567-f006:**
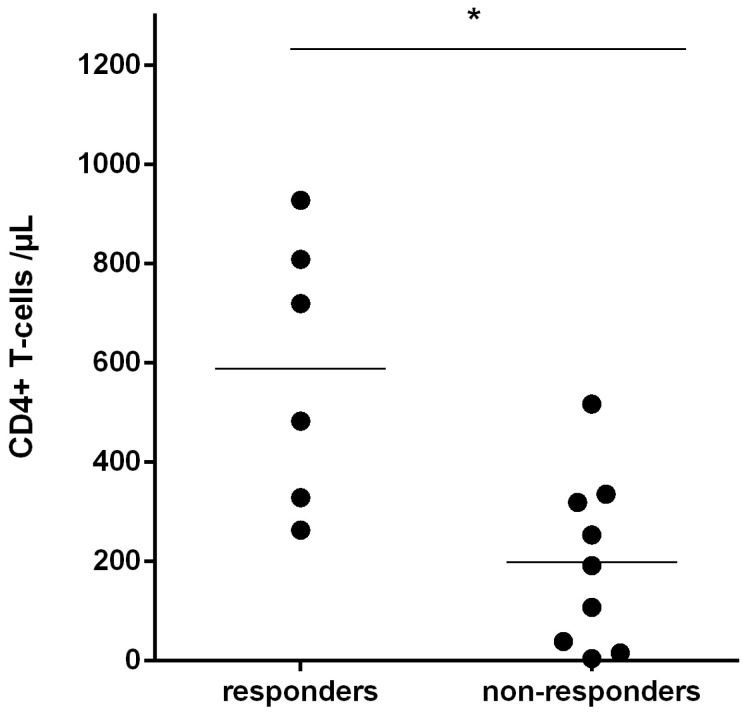
Pre-ART CD4 counts in HLA-B*52+ subjects. HIV-1-infected HLA-B*52+ subjects with detection of Rev-RI8-specific T-cells (responders) displayed significantly higher CD4+ T-cells (median 602/µL, range 263–928) than subjects without recognition of Rev-RI8 (non-responders, median CD4+ T-cells: 192/µ, range 5–517), Mann–Whitney-U-Test: *p* = 0.012 (*p* = 0.0290, if #1 is omitted). For the only untreated subject (controller #1) the CD4 count at the time point of ELISpot analysis of Figure 1a was used (CD4 928/µL). Significance with *p* < 0.05 is indicated by *.

**Table 1 viruses-15-00567-t001:** Immunological and virological parameters in controller #1.

Days after Diagnosis	CD4/µL	CD8/µL	Viral Load in Copies/mL
0	1004	466	40
4	n.d. ^1^	n.d. ^1^	<20
14	857	529	20
125	977	553	20
204	1074	547	20
293	863	325	20
383	928	439	250
453	1184	592	<20

^1^ n.d. = not done.

## Data Availability

The data presented in this study are available on request from the corresponding author.

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
