# Peer review of "Definition of a New HLA B*52-Restricted Rev CTL Epitope Targeted by an HIV-1-Infected Controller"

_viruses, 2023, doi:10.3390/v15020567_

Round 1
Reviewer 1 Report
This is an interesting study which uses well-established methods to characterise in detail the HIV-specific T-cell response of a donor presenting with spontaneous viral control (without ART). In addition to previously defined responses, including responses restricted by HLA-B57, which is strongly associated with viral control, the authors define a new epitope in HIV-1 rev. They go on to show that this response is restricted by HLA-B52 and is also found in other donors with this allele (also associated with viral control). Importantly they show that B52 donors with this response have a higher CD4 count at presentation than those without. The authors go on to describe in the Discussion how the epitope lies in a functionally important part of the rev protein.
Overall this is an interesting and well-written report. The experimental work has been carefully performed and the conclusions are justified by the results. I have only one minor point: it would be useful to have a table showing the recorded variation in this region of rev, and in particular how this varies between HIV subtypes.
Author Response
We would like to thank the reviewer for the time and efforts to review our manuscript. As suggested we have added two supplementary tables (Tables S3 and S4) with information regarding the sequence variation within the Rev-RI8 epitope in the most frequent HIV-1 subtypes based on the published Rev sequences in the Los Alamos HIV Sequence compendium 2019.
Reviewer 2 Report
In this article, Thomas Harrer and colleagues investigate T-cells specific anti-HIV-1 response in HIV-1 infected elite controller. Using IFN gamma ELISpot, the authors found strong polyclonal T cells response to epitopes in Gag, Nef and Rev. The predominant response was found to include HLA-B*57 epitope AISPRTLNAW and Rev-derived epitope RQRQIRSI (Rev-R18). The R-18 epitope was not previously described, Its functionality was confirmed by the analysis of T cell responses in a cohort of HIV-1 infected HLA-B*52 positive individuals. Of 15 participants tested, 6 individuals showed recognition of Rev R-18 epitope. Also, those that recognized the epitope had higher CD4 levels prior to the start of antiviral therapy. Overall, this is well presented and clear study. The only remaining question is how conserved is the R18 epitope? There is also a suggestion to improve Figure 4 as detailed below.
Major critiques:
1. Please analyze conservation of RQRQIRSI sequence using publicly available HIV-1 sequences.
2. I suggest redoing Figure 4, by adding an additional point (i.e., 3 ug/ml), plotting x-axis in log and calculating IC50 by non-linear fit.
Author Response
We would like to thank the reviewer for the time and efforts to review our manuscript.
We have addressed the points of critiques as follows:
1: "Please analyze conservation of RQRQIRSI sequence using publicly available HIV-sequences."
As requested we have added two supplementary tables (Tables S3 and S4) with information regarding the sequence variation within the Rev-RI8 epitope in the most frequent HIV-1 subtypes based on the published Rev sequences in the Los Alamos HIV Sequence compendium 2019. In table S3 we present the published Rev-RI8 sequences for each HIV subtype. In table S4, we analyze the frequency of amino acid substitutions for each amino acid position in the different HIV subtypes. We have updated the corresponding reference [19] (HIV sequence database 2019 instead of 2018).
2. "I suggest redoing Figure 4, by adding an additional point (i.e., 3 ug/ml), plotting x-axis in log and calculating IC50 by non-linear fit."
As suggested, we have modified Figure 4 by plotting the x-axis in log and calculating the 50% peptide sensitizing activity by non-linear fit. We have added the 50% peptide sensitizing concentrations of the three different experiments with the three different CTL lines in the legend of figure 4 and calculated the mean 50% EC50 concentration. In addition, we have added the mean 50% sensitizing concentration within the text (line 272).
We could not add an additional data point at a 3 µg/ml concentration as we used only log peptide dilutions in our experiments and had not measured at the 3 µg/ml concentration. The non-linear fit calculations clearly indicated that the measured concentrations were valid to calculate the 50% peptide activity concentration (EC50). This is underlined by the similar EC50 values obtained by three different experiments with three different CTL lines. To add new additional data points we would have to obtain new blood samples from the patients and to perform new experiments with ex vivo generation of Rev-RI8-specific CTL lines. If this is required, we need an extension of the revision time by 2 months as the patients come only every 3-4 months to their regular visits.
Round 2
Reviewer 2 Report
All my concerns were answered